# The *KRAS*-Mutant Consensus Molecular Subtype 3 Reveals an Immunosuppressive Tumor Microenvironment in Colorectal Cancer

**DOI:** 10.3390/cancers15041098

**Published:** 2023-02-08

**Authors:** Pariyada Tanjak, Amphun Chaiboonchoe, Tharathorn Suwatthanarak, Onchira Acharayothin, Kullanist Thanormjit, Jantappapa Chanthercrob, Thanawat Suwatthanarak, Bundit Wannasuphaphol, Kemmapon Chumchuen, Bhoom Suktitipat, Somponnat Sampattavanich, Krittiya Korphaisarn, Ananya Pongpaibul, Naravat Poungvarin, Harald Grove, Woramin Riansuwan, Atthaphorn Trakarnsanga, Asada Methasate, Manop Pithukpakorn, Vitoon Chinswangwatanakul

**Affiliations:** 1Department of Surgery, Faculty of Medicine Siriraj Hospital, Mahidol University, Wanglang Road, Bangkok 10700, Thailand; 2Siriraj Cancer Center, Faculty of Medicine Siriraj Hospital, Mahidol University, Bangkok 10700, Thailand; 3Siriraj Center of Research Excellent for Systems Pharmacology, Faculty of Medicine Siriraj Hospital, Mahidol University, Bangkok 10700, Thailand; 4Siriraj Genomics, Faculty of Medicine Siriraj Hospital, Mahidol University, Bangkok 10700, Thailand; 5Department of Biochemistry, Faculty of Medicine Siriraj Hospital, Mahidol University, Bangkok 10700, Thailand; 6Integrative Computational Bioscience Center, Mahidol University, Nakhon Pathom 73170, Thailand; 7Division of Medical Bioinformatics, Research Department, Faculty of Medicine Siriraj Hospital, Mahidol University, Bangkok 10700, Thailand; 8Department of Medicine, Faculty of Medicine Siriraj Hospital, Mahidol University, Bangkok 10700, Thailand; 9Department of Pathology, Faculty of Medicine Siriraj Hospital, Mahidol University, Bangkok 10700, Thailand; 10Department of Clinical Pathology, Faculty of Medicine Siriraj Hospital, Mahidol University, Bangkok 10700, Thailand; 11Division of Medical Genetics, Department of Medicine, Faculty of Medicine Siriraj Hospital, Mahidol University, Bangkok 10700, Thailand

**Keywords:** *KRAS* mutation, colorectal cancer, tumor microenvironment, immunosuppressive tumor microenvironment, TGFβ signaling

## Abstract

**Simple Summary:**

The poor prognosis outcome of patients with *KRAS* mutations (*KRAS*^mut^) was correlated with an immunosuppressive tumor microenvironment (TME). At the gene expression level and pathway analysis, *KRAS*^mut^ tumor activates TGFβ signaling to reduced proinflammatory and cytokine gene signatures. Spatial profiling in the TME region of *KRAS*^mut^, classified as consensus molecular subtype 3 (CMS3), showed an up-regulation of genes *CD40*, *CTLA4*, *ARG1*, *STAT3*, *IDO*, and *CD274*, associated with immunosuppression in TME.

**Abstract:**

Colorectal cancers (CRC) with *KRAS* mutations (*KRAS*^mut^) are frequently included in consensus molecular subtype 3 (CMS3) with profound metabolic deregulation. We explored the transcriptomic impact of *KRAS*^mut^, focusing on the tumor microenvironment (TME) and pathways beyond metabolic deregulation. The status of *KRAS*^mut^ in patients with CRC was investigated and overall survival (OS) was compared with wild-type *KRAS* (*KRAS*^wt^). Next, we identified CMS, and further investigated differentially expressed genes (DEG) of *KRAS*^mut^ and distinctive pathways. Lastly, we used spatially resolved gene expression profiling to define the effect of *KRAS*^mut^ in the TME regions of CMS3-classified CRC tissues. CRC patients with *KRAS*^mut^ were mainly enriched in CMS3. Their specific enrichments of immune gene signatures in immunosuppressive TME were associated with worse OS. Activation of TGFβ signaling by *KRAS*^mut^ was related to reduced pro-inflammatory and cytokine gene signatures, leading to suppression of immune infiltration. Digital spatial profiling in TME regions of *KRAS*^mut^ CMS3-classified tissues suggested up-regulated genes, *CD40*, *CTLA4*, *ARG1*, *STAT3*, *IDO*, and *CD274*, that could be characteristic of immune suppression in TME. This study may help to depict the complex transcriptomic profile of *KRAS*^mut^ in immunosuppressive TME. Future studies and clinical trials in CRC patients with *KRAS*^mut^ should consider these transcriptional landscapes.

## 1. Introduction

Colorectal cancer (CRC ) is a heterogenous disease defined by various alterations in several genes and signaling pathways, both within tumor cells and in the tumor microenvironment (TME), driving its progression and invasion [1,2]. Approximately 30% of CRC arises via an activation of the mitrogen-activated protein kinase (MAPK) signaling pathway that is associated with the presence of *KRAS*^mut^ [3]. CRC patients with *KRAS*^mut^ revealed a poor prognosis [4], liver metastases [5], and tumor aggressiveness [6], compared to patients with *KRAS*^wt^. The status of *KRAS*^mut^ is used as an established biomarker of resistance to anti-epidermal growth factor receptor (EGFR) therapies [7,8]. The predictive value of *KRAS*^mut^ in other roles is inconclusive. Due to the intrinsic characteristics of the oncogenic KRAS protein, blocking KRAS has been quite a challenge. Many efforts have focused on indirectly targeting *KRAS*^mut^ tumors, including immunotherapeutic approaches [9]. However, the biological interaction between cancer cells that harbor *KRAS*^mut^ and their surrounding immune cells in the TME of CRC is unclear.

The four consensus molecular subtypes (CMSs) are currently the best description of CRC heterogeneity at the gene expression level. Although *KRAS*^mut^ is enriched in CMS3 with a prominent signature indicating metabolic derangement, *KRAS*^mut^ tumors can be found in all subtypes [10]. Furthermore, *KRAS*^mut^ tumors classified as CMS2 and 3 revealed relatively poor immune infiltration [10,11,12]. It is possible that *KRAS*^mut^ occurs in a different biological context with metabolic alterations in CRC, and may have an additional role in cancer immunology. 

Immune cells are a major cell composition in the TME [13]. Some immune cells often assist cancer cells to thrive and successfully escape immune surveillance [2]. The TGFβ signaling pathway is reported as a key mediator of the *KRAS* mutation in the invasion of CRC by evading the immune system [14,15,16,17]. Up regulation of TGFβ signal is associated with activation of the epithelial to mesenchymal transition (EMT) pathway, which is strongly associated with immune escape in the TME [18]. In the past decade, molecular characteristics related to the prognosis of CRC have focused mainly on cancer cells [19,20]. CMS classification is complemented by transcriptomic analyses of whole tumor samples without compartmentalization between cancer cells and TME. Although the composition of immune and stromal cells has been reported, in which four CMS by using their bulk RNA [13,21,22], the spatial molecular heterogeneity between cancer and TME regions of each CMS has not been investigated. Exploring the interaction between *KRAS*^mut^ and gene expression profiling beyond MAPK and metabolic signaling pathways could provide more information on the biology of CRC and facilitate the discovery of therapeutic targets in *KRAS*^mut^ CRC. Here, we report the differential gene expression profile associated with *KRAS*^mut^, with a focus on the immune TME of *KRAS*^mut^ CRC. 

## 2. Materials and Methods

### 2.1. Patients and Tissue Samples

The cohort included patients aged 18 years and older who had colorectal adenocarcinoma. Written informed consent was obtained from each patient, and the study protocol was approved by the Siriraj Institutional Review Board (certificate of approval number Si156/2011 and Si593/2019). All experiments were performed according to the relevant guidelines and regulations. A total of 97 fresh frozen colorectal adenocarcinoma samples were collected from 97 CRC patients who underwent surgical treatment in the Department of Surgery, Faculty of Medicine Siriraj Hospital, Mahidol University, between October 2010 and March 2011.

### 2.2. Survival, Clinical and Statistical Analysis

Analyses were performed using the R program (version 3.6; R Foundation for Statistical Computing, Vienna, Austria). *t*-test, Chi-squared, or Fisher’s exact tests were performed to test the statistical significance of clinical characteristics. A *p*-value of <0.05 was considered statistically significant. Overall survival (OS), the main endpoint, was determined from the date of diagnosis to the date of death from any cause. The end of the surveillance period was 31 January 2022. The patient’s survival outcome was analyzed using Kaplan–Meier analysis, in the “survminer” R package. Differences between curves were compared using the logarithmic rank test. Cox regression analysis was used to study the association between survival and clinicopathological variables in univariate and multivariate analyses. Hazard ratio (HR) with corresponding 95% confidence intervals (CI) and *p*-value were estimated. 

### 2.3. DNA Isolation

The genomic DNA of all patients’ tissues was extracted from fresh frozen tissue using the QIAamp DNA Mini Kit (Qiagen, Hilden, Germany), according to the manufacturer’s instructions. DNA was extracted from all frozen cancer tissues after an overnight proteinase K digestion step at 65 °C. After extraction, all DNA samples were subjected to RNAse treatment (Qiagen, Hilden, Germany) and optimized in an elution step.

### 2.4. RNA Isolation

All fresh frozen tissues were placed in lysis buffer with lysing matrix Z (MP Biomedicals, Santa Ana, CA, USA) and homogenized using the FastPrep-24™ 5G Instrument (MP Biomedicals, Santa Ana, CA, USA). The RNA was then extracted using the RNeasy Mini Kit (Qiagen, Hilden, Germany), according to the manufacturer’s protocol. RNA integrity was assessed using the RNA Nano 6000 Assay Kit and the Bioanalyzer 2100 system (Agilent Technologies, Santa Clara, CA, USA). 

### 2.5. KRAS Mutation Screening

Mutations in exon 2 (codon 12 and 13) of the *KRAS* gene were analyzed with the therascreen KRAS kit (Qiagen, Hilden, Germany) which used allele-specific amplification achieved by an amplification refractory mutation system, according to the manufacturer’s instructions. All negative samples were subsequently subjected to Sanger sequencing to identify *KRAS*^mut^ in exon 3 (codon 61).

### 2.6. KRAS-Related Gene Expression Profiling Using NanoString Platform

The nCounter^®^ analysis system was used to perform the assay (NanoString Technologies, Seattle, WA, USA). A pancancer progression panel kit was used to measure the expression of 770 genes. The raw counts of each target gene were normalized by the geometric mean counts of 11 housekeeping genes (*HRNP1*, *RPL27*, *RPL9*, *RPL6*, *RPL30*, *OAZ1*, *PTMA*, *RPS29*, *UBC*, *RPS12*, *and RPS16*) and spiked controls. A threshold count value equal to 20 was used for background thresholding and normalizing the samples for differences in hybridization. Raw data were processed into a signature matrix using nSolver Analysis Software version 4.0 (NanoString Technologies Inc., Seattle, DC, USA). The read counts from the raw data output were evaluated for differentially expressed genes (DEG) and cell type scoring, after normalization using ROSALIND software. Gene expression data were normalized using the DESeq2 package in R. Hierarchical clustering and DEG exploration were carried out [23]. The identification of DEG in gene expression between *KRAS*^mut^ and *KRAS*^wt^ was determined using a *t*-test with a significance threshold of *p*-value < 0.05. The Benjamini-Hochberg method was applied to adjust the *p*-value as the false discovery rate (FDR). The graphs were plotted using the ggplot2 and ComplexHeatmap packages in R.

### 2.7. Pathway Analysis

Ingenuity Pathway Analysis Software (IPA 84978992, Ingenuity® Systems, https://digitalinsights.qiagen.com/, accessed on 31 October 2022) was used to examine the biological network associated with *KRAS*^mut^. The IPA software (IPA 84978992) uses a manually curated database that contains information from several reputable sources, including published journal papers and gene annotation databases. Fisher’s exact test was used to calculate the probabilities between the input gene set and the canonical pathway, disease, and function. IPA also predicted the upstream and downstream effects of activation or inhibition on other molecules on the basis of the expression data from the input gene set.

### 2.8. CMS Classification Using the NanoString Platform

CRC subtype classification based on deep learning, or DeepCC, a supervised functional spectra-based cancer subtyping stratification model, was implemented to identify CMS from the CRC gene expression profile [24]. A gene expression data set of Siriraj hospital’s CRC cohort was logarithm transformed and converted from genetic information to functional spectra associated with biological pathway activities. Subsequently, a DeepCC model (DeepCC R package version 0.1.1), containing a trained artificial neural network, was performed to extract advantageous features and classify the Siriraj hospital gene expression data into four CMS classes, CMS1, CMS2, CMS3, and CMS4 [10].

### 2.9. Digital Spatial Profiling (DSP)

Formalin-fixed, paraffin-embedded tissue slides (FFPE) from seven patients (*KRAS*^mut^, *n* = 4; *KRAS*^wt^, *n* = 3), from a subset of 97 patients, were strictly prepared for DSP using manual instruction from the GeoMx instrument and the GeoMx immune pathway panel kit with 84 genes (NanoString Technologies Inc.). A tricolor panel of fluorescence morphology markers was used, targeting Pan-cytokeratin (Pan-CK, epithelial and tumoral regions), CD45 (immune cells), and SYTO13 (nuclear) on stained slides. Slides were loaded onto the GeoMx instrument, scanned, and selected for regions of interest (ROI) for 58 ROIs. To ensure reliable quantification and comparison of inter-ROI data, ROI surface areas were drawn between 202,791–340,192 μm^2^, encompassing between 498–548 nuclei. The raw data were then counted in the nCounter analysis system using standard procedures. Raw digital count files (RCC) for individual ROIs were normalized by the geometric mean of the housekeeping genes *RAB7A*, *OAZ1*, *UBB*, *POLR2A*, and *SDHA*. Normalized data were logarithmically transformed with or without being median centered prior to comparison and plotting. All data were processed and analyzed in DSP analysis software. Volcano plots were created with a log_2_ fold change and an adjusted *p*-value at 0.05 for cut-off. For differential expression analysis, a non-parametric Mann–Whitney U test with a *p*-value < 0.05 at the significant cutoff was conducted. 

## 3. Results

### 3.1. CRC Patients with KRAS^mut^ Show a Shorter Overall Survival That Is Mainly Correlated with Their Transcriptomic Profile

To study the effect of *KRAS*^mut^ on gene expression level, we initially investigated patient characteristics and *KRAS*^mut^ status in our cohort *(n* = 97). Univariate analyses of clinicopathologic characteristics according to the *KRAS*^mut^ status are summarized in Table 1. We observed *KRAS*^mut^ in 41.24% (*n* = 40) of the cases. Mutations occurred mainly in exon 2 (total *n* = 36; codon 12, *n* = 28; codon 13, *n* = 8) and rarely in exon 3, codon 61 (*n* = 4). The baseline clinical characteristics of all patients were similar between the *KRAS*^wt^ and *KRAS*^mut^ groups without being statistically different. 

Like previously published data, patients with *KRAS*^mut^ had a worse clinical outcome (Figure 1A). Based on tissue samples and RNA quality of each sample, different numbers of samples were available for the various assays. We found fresh frozen specimens of patients with *KRAS*^wt^ (*n* = 25) and *KRAS*^mut^ (*n* = 34) met the RNA quality to study their gene expression profile using the NanoString platform. Notably, between patients (*n* = 59) with *KRAS*^wt^ and *KRAS*^mut^, we still observed significant overall survival (Figure 1B). This implied that the transcriptomic profiling study of 34 patients with *KRAS*^mut^ was associated with the worst outcome.

### 3.2. DEG of KRAS^mut^ Tumors Enrich in Immune Signature and TGFβ Pathways

After RNA quality assessment, we examined gene expression and classified the CMS of the patients (*n* = 59) using their gene expression profile from the NanoString platform. Although the *KRAS*^mut^ tumor could be assigned from CMS1 to CMS4, most of the *KRAS*^mut^ tumors were clustered in CMS3 (Figure 2A and Appendix A). We found 92 (53 increased and 39 decreased) nominally significant DEGs of *KRAS*^mut^ with a *p*-value < 0.05, which *FREM1*, *ERMP1* (up-regulated), and *CCL8* (down-regulated) were significant after multiple test corrections (FDR < 0.05). After adjustment for age and sex, *CCL8* still remained statistically differentially expressed, with levels lower in the *KRAS*^mut^ group (*p*-value = 0.01). To gain insight into the pathways involved, we used gene set analysis to determine which nominal DEGs had been annotated or identified using ROSALIND and nCounter software. The analysis of the gene set showed that DEG is enriched in the top-five pathways (*p*-value < 0.01) in the sources of the BioPlanet and WikiPathways database sources (Figure 2B). Surprisingly, we found that DEGs were mainly enriched in immune pathways in both databases, such as the immune system pathway (BioPlanet) and the innate immune system (BioPlanet), as well as interactions between immune cells and microRNAs in the TME pathway (WikiPathways). 

To identify significant canonical pathways (*p*-value < 0.001) in 92 genes up- or down-regulated, IPA was used. As expected, the HIF1α signaling pathway, which normally involves metabolic reprogramming, was also included in the IPA graphical summary. It can be implied that *KRAS*^mut^ reprogramed the metabolic pathway via the HIF1α signaling pathway. In particular, the IPA graphical summary suggested the roles of *CCL11* and *TNFSF12* in down-regulating cell movement and cellular infiltration of lymphocytes (Figure 2C). Thus, we focused on the immune and TME pathways using the canonical pathway analysis. We found that *KRAS*^mut^ DEGs were also enriched in the regulation of EMT, macrophage stimulating protein receptor d’origine nantais (MSP-RON), and TGFβ signaling pathways with *p*-value < 0.001 (Appendix A). These signaling pathways have been reported to associate with maintaining the stability of TME and contributing to immune escape in immune TME [18,25].

We then investigated the upstream regulator analysis of the IPA. Interestingly, the result revealed that *TGFBR1* played a regulatory key role for *STAT1*, *CXCL10*, *BMP4,* and *CDKN1A* in the inhibition of lymphocyte cell death, which to be T-regs (Figure 2D and Appendix A). Taken together, these findings suggested that the transcriptomic profile of *KRAS*^mut^ is not only linked to CMS3-enrichement, a prominent metabolic adaptation in CRC at the pathway level, but also enables remodeling of immune TME via activation of the TGFβ signaling pathway to reduce pro-inflammatory cytokines and suppress immune infiltration.

### 3.3. KRAS^mut^ CMS3 Classified Tumors Show a Distinct Immune Suppression of the Gene Expression Pattern in the TME 

To decipher the molecular changes of the TME in *KRAS*^mut^ at the gene expression and pathway levels, we used DSP GeoMx technology to profile different gene expression between TME in *KRAS*^mut^ and TME in *KRAS*^wt^. From our last result, *KRAS*^mut^ was mainly encompassed in CMS3, therefore, we addressed the heterogeneity of CMS3 in terms of *KRAS* mutational status and focused on transcriptomic profiling in TME regions. We selected samples from four patients with *KRAS*^mut^ and three patients with *KRAS*^wt^, which were all classified as CMS3 (Figure 3A). The seven samples were a subset of 97 patients. CMS classification of seven patients was performed by using their gene expression profile in dataset GSE220148 from the Gene Expression Omnibus (GEO, https://www.ncbi.nlm.nih.gov/geo/, accessed on 8 December 2022) repository. We found that the expression profiles of 84 genes between the cancer and TME regions showed region-specific expression patterns (Figure 3B). 

Next, we analyzed differential gene expression between the TME regions of *KRAS*^mut^ and the TME regions of *KRAS*^wt^. As shown in Figure 3C, we found 15 significant DEGs for the TME regions of *KRAS*^mut^ with a *p*-value < 0.05 (14 increased and 1 decreased). Interestingly, we observed that only *HLA-DRB* was significantly over-expressed in the TME regions of *KRAS*^wt^. We found significant upregulation of 14 genes (*CD40*, *STAT3*, *CD4*, *IDO1*, *CTLA4*, *CD40LG*, *CXCL10*, *CXCR6*, *CXCL9*, *CD27*, *CD274*, *MSA4*, *YSIR*, and *ARG1*) in the TME regions of *KRAS*^mut^. We also found 37 significant DEGs for cancer regions of *KRAS*^mut^ (Appendix A). 

### 3.4. The upregulation of CD40 in the TME of KRAS^mut^

To determine the critical biological processes and molecular pathway, significant DEGs of the TME regions of *KRAS*^mut^ were imported into IPA. The IPA pathway analysis of DEGs for the TME regions in *KRAS*^mut^ showed that their DEGs were enriched in the TME, primary immunodeficiency and CD40 signaling pathways with a *p*-value < 0.001 (Appendix A). We then focused on identifying the interaction among candidate genes, proteins, and functional roles in which IPA mapped the TME pathway. We found that the TME pathway map showed upregulation of genes, *CTLA4*, *ARG1*, *STAT3*, *IDO*, and *CD274*, and was associated with a reduction of immune infiltration of cytotoxic T cell lymphocytes in TME regions (Appendix A). However, these genes were likely related to the upregulation of genes in the cancer regions of *KRAS*^mut^. In order to identify the molecular differences between TME of *KRAS*^mut^ and *KRAS*^wt^, we determined the contrasting upregulation of DEGs between TME and cancer regions in each *KRAS* tumor type (Figure 4A and Appendix A). We found *CD40* was the up-regulated gene in TME of *KRAS*^mut^. Taken together, the IPA analyses of DEG for *KRAS*^mut^ from NanoString and DEGs for TME of *KRAS*^mut^ from GeoMx suggest that *KRAS*^mut^ tumors suppress immune infiltration by activating the TGFβ signaling pathway. In *KRAS*^mut^ CMS3 classified tumors, overexpression of *CD40* was associated with immunosuppressive signals, resulting in the upregulation of *CTLA4*, *ARG1*, *STAT3*, *IDO*, and *CD274* in TME (Figure 4B).

## 4. Discussion

CRC is a heterogeneous disease that involves multiple genes and signaling pathways to drive cancer progression [1]. The surrounding TME also strongly communicates with cancer cells to support cancer growth, cancer evasion, and influence their response to therapy [26,27,28]. Several studies point out that mutant *KRAS* influences on the composition of the immune microenvironment through multiple mechanisms [9,11]. CMS3 is enriched in tumors with *KRAS*^mut^ and shows an adaptation to the metabolic pathway, however, other roles of *KRAS* remain unclear.

Here, we observed that the gene profile of *KRAS*^mut^ tumors was classified mainly as CMS3 that showed metabolic adaptation through HIF1α signaling. Importantly, we found that *KRAS*^mut^ DEG was enriched in immune pathways and led to an immunosuppressive environment in TME. Our finding suggested that this phenomenon occurred through an activation of TGFβ to reduce proinflammatory and cytokine gene signatures such as *CCL8*, *CCL11*, *CXCL10*, and *TNFSF12* leading to immune suppression in patients with CRC. The gene expression profile in CMS3 TME of *KRAS*^mut^ shows upregulation of *CD40*, *CTLA4*, *ARG1*, *STAT3*, *IDO*, and *CD274*, making it a key regulator of immune suppression in TME regions surrounding the *KRAS*^mut^ tumor. Our data might help to uncover the complex interrelationship among gene expression, pathway, and function of *KRAS*^mut^ tumors leading to immunosuppressive TME.

The EMT signaling pathway was regulated by TGFβ signaling, which plays a key role in the progression of colorectal cancer by evading the immune system [15,29,30]. TGFβ signaling plays a critical role in suppressing immunity in the TME by altering T-regs [31]. Several studies have shown that TGFβ can induced many genes of stromal cells in TME, contributing to cytokine and chemokine secretion [32,33,34,35,36,37]. Consistent with these findings, we found that TGFβ signaling at gene expression level is associated with *KRAS*^mut^. We suggested that upregulation of *TGFBR1* reduced proinflammatory and cytokine gene signatures such as *CCL8*, *CCL11*, *CXCL10*, and *TNFSF12* leading to inhibition of T-reg cell death in CRC patients.

CMS3 is highly enriched for patients with *KRAS*^mut^, which is characterized by poor infiltration of immune cells [13], however, CMS3 also includes a group of patients with *KRAS*^wt^. We found that *KRAS*^mut^ CMS3 is most likely to represent the tumor-associated immunosuppressive microenvironment subtype. To explain the association of *KRAS*^mut^ CMS3 in immunosuppressive TME, digital spatial profiling was conducted. The use of digital spatial profiling to address the question of molecular targets and pathways at the RNA level might help identify indirect target oncogenic *KRAS* in CMS3. We investigated transcriptomic profiling in the TME regions, comparing *KRAS*^mut^ and *KRAS*^wt^ of CMS3, focusing on the IPA TME pathway map. Our results suggested that DEGs for TME of *KRAS*^mut^ tumors increased the gene expression of *CTLA4*, *ARG1*, *STAT3*, *IDO*, and *CD274* to shape immunosuppressive TME. 

These genes have been reported at various stages of cancer progression such as angiogenesis, anti-apoptosis, and immune evasion [38,39,40,41,42,43]. Accumulating evidence indicates that the *KRAS*^mut^ tumor invasion mechanism plays an important role in suppressing immune cell responses within TME by recruiting immunosuppressive cells, including MDSC, and T-regs. However, in *KRAS*^mut^ tumors that intersect with a CMS classification reveal a low expression of *CTLA4* in *KRAS*^mut^ tumors at RNA level [10,12]. These contrasts may be explained by DEGs for TME regions of *KRAS*^mut^ enriched in primary immunodeficiency signaling pathways, therefore, upregulation of the *CTLA4* probably represents high T-regs in TME. It should be noted that the expression of inhibitory markers is not always a sign of the immune response, but it may also be evidence of an immune deficiency pathway that is a major obstacle to the antitumor immune response. The use of spatial profiling that is specified for the TME might enhance precise treatment, especially in immunotherapy. In the TME regions of *KRAS*^mut^ tumors, we further found significant downregulation of *HLA-DRB*. *HLA-DRB* overexpression is strongly related to a better prognosis in CRC patients [44,45]. Together, *HLA-DRB* overexpression in TME could be used as a biomarker with a good prognosis of *KRAS*^wt^ classified as CMS3. 

In addition to recruiting immunosuppressive cells, MDSC, tumor-associated macrophages, and tumor-associated neutrophils, play an important role in tumor progression and immunosuppressive function [46,47,48]. Tumor-associated neutrophils exhibit immunosuppressive function by producing ARG1 and IDO [49]. Furthermore, tumor-associated macrophages exert an immunosuppressive effect by producing the chemokines and NF-κB p65/STAT3, which preferentially recruit non-cytotoxic T cell subsets [50]. Therefore, upregulation of *STAT3*, *ARG1* and *IDO* in TME of *KRAS*^mut^ tumors in our results might be due to the infiltration of tumor-associated macrophages and tumor-associated neutrophils in TME. Moreover, our result showed the upregulation of *CD40* in TME of *KRAS*^mut^ tumors, which is in contrast to the TME of *KRAS*^wt^. As a member of tumor necrosis factor receptor superfamily, CD40 is most prominently expressed on dendritic, myeloid, and B cells [51]. Agonistic CD40 antibodies have been explored with anti-angiogenic agents in the experimental model, which show that they improved tumor infiltration by cytotoxic T cells [52]. We therefore hypothesized that *CD40* activation might not only improve cytotoxic T cell infiltration but also enhance immunosuppressive cells in the TME. Taken together, our results suggested that upregulation of *CD40*, *CTLA4*, *ARG1*, *STAT3*, *IDO*, and *CD274*, which were included in tumor inflammation signature [53], in the TME of *KRAS*^mut^, could be characteristic of immune suppression.

Despite this, our transcriptomic study had some limitations. First, mutation testing was limited to *KRAS* exon 2 (codons 12 and 13) and exon 3 using classic real-time PCR and Sanger sequencing, which may have low sensitivity compared to modern next-generation sequencing techniques. Second, our study cohort may be small in size; however, ours includes long-term follow-up data that still showed the statistical significance of survival between *KRAS*^mut^ and *KRAS*^wt^ patients. Third, although the immunogenic transcriptomic landscape of *KRAS*^mut^ in this study suggested some targeted genes, such as *TGFBR1*, *CD40*, *CTLA4*, *ARG1*, *STAT3*, *IDO*, and *CD274*, their functional proteins need to be confirmed. 

## 5. Conclusions

In summary, our study has studied the clinically significant genomic transcriptomic and heterogeneity of the TME in CMS3 CRC patients with *KRAS*^mut^ compared to *KRAS*^wt^. Overall, the data presented here reveal that *KRAS*^mut^ tumor activates TGFβ signaling to reduced proinflammatory and cytokine gene signatures, resulting in recruitment of immunosuppressive cells, including MSDS and T-reg in the TME. Our data might help to explain the complex transcriptomic interrelationship in TME of *KRAS*^mut^ in terms of immune suppression. Future studies and clinical trials in *KRAS*^mut^ CRC should consider these transcriptomic findings. 

## Figures and Tables

**Figure 1 cancers-15-01098-f001:**
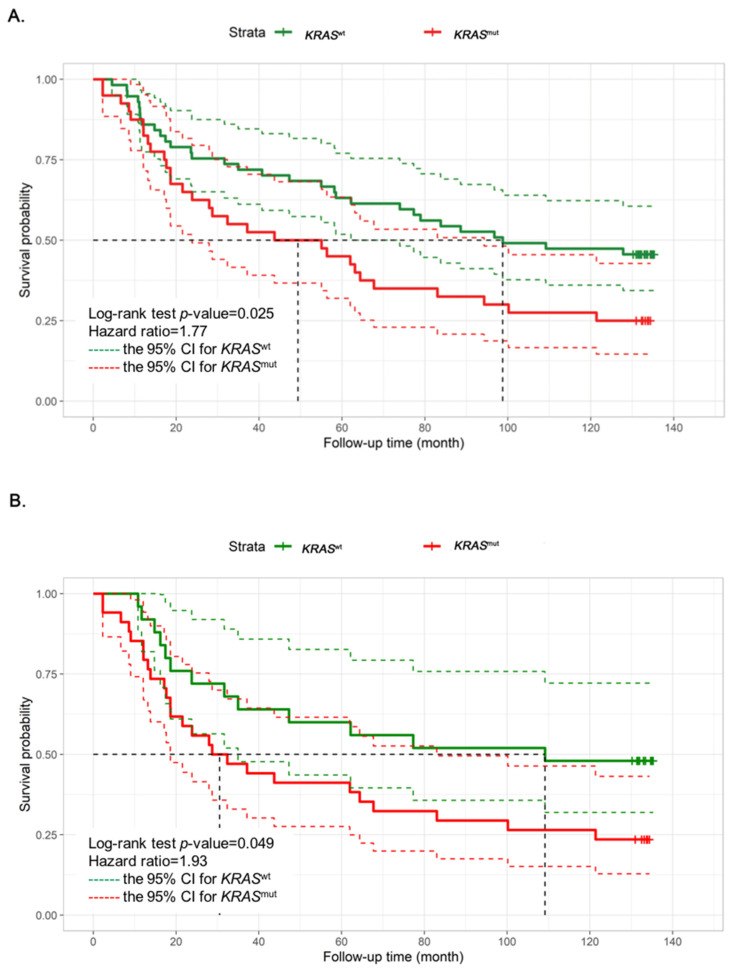
Kaplan–Meier graphs depicting overall survival (OS) of colorectal cancer patients according to the mutational status of *KRAS* in (**A**) all cohort (*n* = 97) and (**B**) transcriptomic study cohort (*n* = 59). Each graph shows univariate hazard ratios (95%CI) and log rank *p*-values. The dashed lines indicate the 95% CI for *KRAS*^wt^ (in green dash) and *KRAS*^mut^ (in red dash). *KRAS*^wt^, *KRAS* wild type; *KRAS*^mut^, *KRAS* mutation; CI, confidence interval.

**Figure 2 cancers-15-01098-f002:**
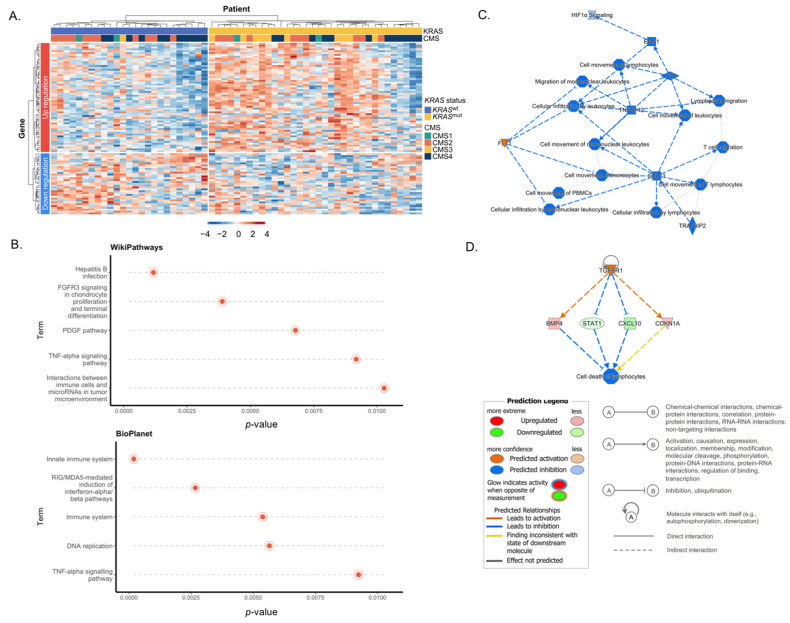
Differential gene expression between *KRAS*^mut^ and *KRAS*^wt^ with a *p*-value < 0.05. (**A**) Heatmap for 92 DEGs of *KRAS*^mut^ (rows) from the NanoString PanCancer progression panel are clustered by statistically significant downregulation (blue) and up regulation (red) between *KRAS*^mut^ and *KRAS*^wt^ (column). The consensus molecular subtypes 1–4 (CMS1–4) were also clustered (column). Blue to red denotes gene expression, with blue implying low gene expression and red implying high gene expression. (**B**) Top five pathways of BioPlanet and WikiPathways database sources enriched by DEGs with their FDR values. (**C**) The IPA graphical summary of 92 statistically significant DEG of *KRAS*^mut^. (**D**) The IPA upstream regulator analysis reveal that *TGFBR1* is the upstream regulator associated with lymphocyte cell death.

**Figure 3 cancers-15-01098-f003:**
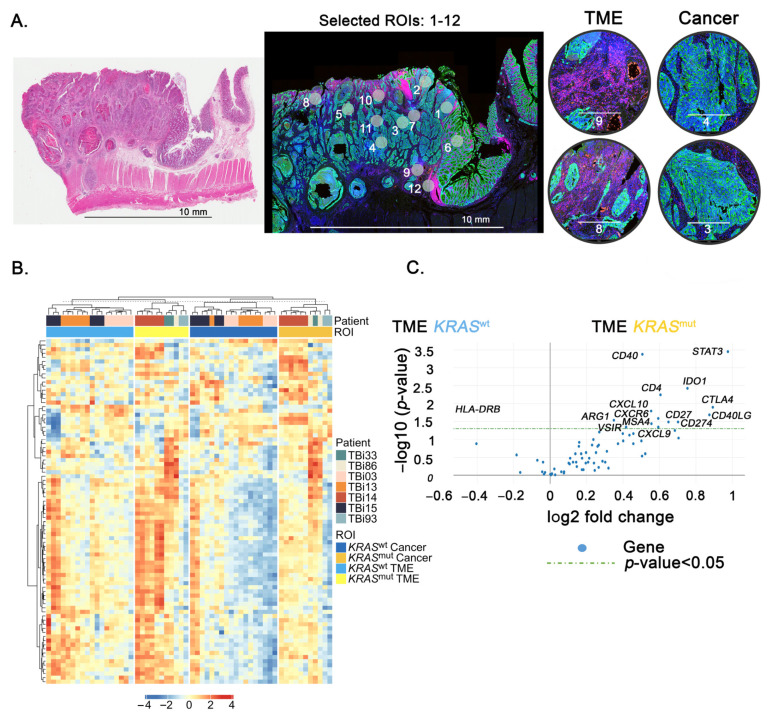
Differential gene expression in TME or cancer regions of patients with CRC *KRAS*^mut^ or *KRAS*^wt^ from DSP analysis. (**A**) Representative images of the TBI03 CRC sample stained by hematoxylin and eosin (H&E) (**left**) and immunofluorescence with selected ROIs: 1-12 (**middle**), cancer or TME ROIs (**right**). Representative of ROIs in all patients are shown with tricolor fluorescence labeling (blue: SYTO13; green: Pan-CK; red: CD45 with scale bar 500 μM. (**B**) Heatmap for 84 DEGs (rows) from the GeoMx immune pathway panel are clustered by TME or cancer (column) in *KRAS*^mut^ (yellow) and *KRAS*^wt^ (blue). Blue to red denotes gene expression, with blue implying low gene expression and red implying high gene expression. (**C**) Volcano plot of 84 genes comparing the differential gene expression of TME region between *KRAS*^mut^ and *KRAS*^wt^. The *x*-axis is the log_2_ fold change of gene expression between *KRAS*^mut^ and *KRAS*^wt^. The *y*-axis is the −log_10_ *p* value results. Genes of interest have been annotated within the plot. The blue dots above the green line are genes that are statistically significant upregulation in *KRAS*^mut^ and *KRAS*^wt^.

**Figure 4 cancers-15-01098-f004:**
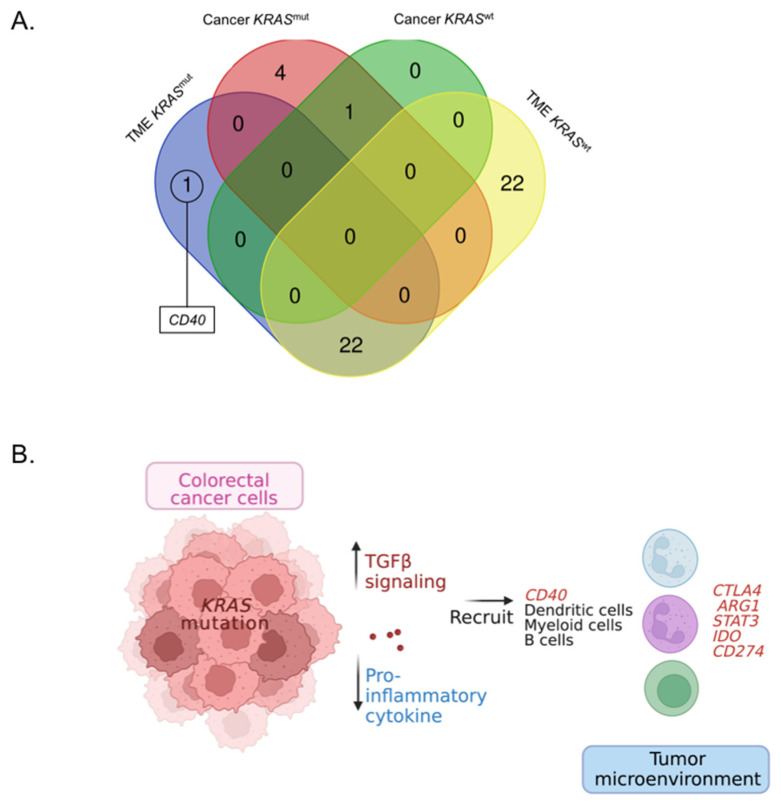
Diagrams illustrating the gene expression patterns in TME of *KRAS*^mut^ (**A**) Venn diagram showing the contrasting *CD40* upregulation in TME of *KRAS*^mut^ (*p*-value < 0.05). (**B**) Diagrams of the immunosuppressive tumor microenvironment of *KRAS*^mut^. Colorectal cancer cells with *KRAS*^mut^ up-regulate TGFβ signaling and down-regulate proinflammatory cytokine to recruit immunosuppressive cells, including myeloid, dendritic, and B cells, which mostly express *CD40*. These cells may up-regulate genes *CTLA4*, *ARG1*, *STAT3*, *IDO*, and *CD274* to suppress the immune function and infiltration of effector cytotoxic T cells in TME.

**Table 1 cancers-15-01098-t001:** Clinical characteristics and *KRAS*^mut^ status of patients with CRC.

Characteristics	Total *N* (%), *n* = 97 (100)	*KRAS*^wt^ *N* (%), *n* = 57 (58.76)	*KRAS*^mut^ *N* (%), *n* = 40 (41.24)	*p*-Value
Mutation status				
Exon 2 codon 12		-	28 (70)	
Exon 2 codon 13		-	8 (20)	
Exon 3 codon 61		-	4 (10)	
Mean age	64.86	63.01	67.48	0.101
Sex				0.459
Male	48	30 (62.5)	18 (37.5)	
Female	49	27 (55.1)	22 (44.9)	
Tumor site				0.381
Rectum	38	18 (50.0)	18 (50.0)	
Left side colon	43	28 (65.1)	15 (34.9)	
Right side colon	18	11 (61.1)	7 (38.9)	
Stage				0.109
I	13	11 (84.6)	2 (15.4)	
II	31	16 (51.6)	15 (48.4)	
III	33	21 (63.6)	12 (36.4)	
IV	20	9 (45.0)	11 (55.0)	
Metastatic site				1.000
Liver	16	8 (50.0)	8 (50.0)	
Lung	1	-	1 (100)	
Liver and lung	2	-	2 (100)	
Others	1	1 (100)	-	
Pathological grade				0.126
Moderately differentiated	86	53 (61.6)	33 (38.4)	
Well differentiated	9	3 (33.8)	6 (66.7)	
Poorly differentiated	2	1(50.0)	1(50.0)	
Adjuvant chemotherapy				0.662
Yes	57	29 (50.9)	28 (49.1)	
No	40	23 (50.9)	17 (49.1)	

*KRAS*^wt^, *KRAS* wild type; *KRAS*^mut^, *KRAS* mutation. *t*-test, Chi-squared or Fisher’s exact tests were performed to test the statistical significance of clinical characteristics.

## Data Availability

Appendix A are attached in additional files. The other datasets used and/or analyzed during the current study are available from the corresponding author on reasonable request.

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
