# Peer review of "The KRAS-Mutant Consensus Molecular Subtype 3 Reveals an Immunosuppressive Tumor Microenvironment in Colorectal Cancer"

_cancers, 2023, doi:10.3390/cancers15041098_

Round 1

Reviewer 1 Report

Tanjak et al. have analyzed the role of KRAS mutation in colorectal cancer and tumor microenviroment. The work is sound, but I want to address some concerns to the authors.

Major concerns

1)    It is not clear how many samples have been analyzed. In material and methods, it is stated that 97 samples were collected (lines 96-97); and 7 for the digital spatial profiling. Are those 7 a subset of the 97 or are different individuals? In addition, it is stated in the results that 59 were analyzed (lines 200 and 212). Why is there this discrepancy?

2)    The differential expression analyses have not been adjusted (for example, by sex and age), have you considered the use of adjustments?

3)    I think that this statement should be corrected: “We identified 92 (53 increased and 39 decreased) significant DEG of KRASmut with a P value<0.05, which FREM1, ERMP1 (upregulated) and CCL8 (downregulated) were expressed significantly differently (FDR <0.05)” (lines 214-216). “Significant DEG” and “expressed significantly differently” is the same. 92 genes were nominally significant, while FREM1, ERMP1 and CCL8 were significant after multiple test correction. And this is a key point: since you have tested 770 genes at the same time, you should correct for multiple test. Then, here we have an issue: with only 3 genes actually significant, it does not make sense to do an enrichment analysis. Or, you should justify using the nominally significant genes. In this case, the results showed in 2B, although it is difficult to read, it seems clear that are not significant, since the X-axis is higher than 0.05.

4)    On the whole, the results are not robust enough to support strong claims like “This study elucidated”, “we demonstrated”, “our study has highlighted”, etc (in abstract, discussion and conclusion). Please, revise those sections to lower the claims.

Minor concerns

1)    In results, differentially expressed genes have been explained in two different sections (2.6 and 2.9). In which results have been used each one?

2)    Figures (in the main text and supplementary material) are difficult to see, please provide high quality images.

3)    There is not supplemental table S3 (line 292)

Author Response

We thank the reviewer 1 for critically reviewing our manuscript. We are grateful for the reviewers’ constructive and important comments/suggestions to improve the quality of this manuscript. Overall, we have made an attempt to revise the manuscript in accordance to the suggestions of the reviewers. All changes in the manuscript has been shown by “Track Changes” function. We trust that the revised manuscript has been improved to the level of the reviewers’ satisfaction in the Cancers. The point-by-point responses to the reviewer 1’s comments/suggestions/queries are as follows:

Major concerns

Point 1: It is not clear how many samples have been analyzed. In material and methods, it is stated that 97 samples were collected (lines 96-97); and 7 for the digital spatial profiling. Are those 7 a subset of the 97 or are different individuals? In addition, it is stated in the results that 59 were analyzed (lines 200 and 212). Why is there this discrepancy?

Response 1: The author would like to agologise for this confusion. We collected the 97 fresh frozen samples from 97 individual patients, however, the RNA quality limited the number of samples in various assays. After RNA quality assessment, we examined gene expression and classified CMS of 59 samples from 59 patients using their gene expression profiling from the NanoString platform. The seven individual samples for the digital spatial profiling (DSP) was also included in 97 samples. Their RNA quality numbers reached to run RNA sequencing, therefore, CMS classification of seven samples ( 7 patients) for DSP using - the RNA sequencing data, which is aviable the Gene Expression Omnibus (GEO, https://www.ncbi.nlm.nih.gov/geo/) repository with an accession number of GSE220148.

We have revised the sentence in material and method (Lines 139 and 211).

Line 139 “ A total of 97 fresh frozen colorectal adenocarcinoma samples were collected from 97 CRC patients who underwent surgical treatment in the Department of Surgery, Faculty of Medicine Siriraj Hospital, Mahidol University between October 2010 and March 2011.”

Line 211 “Formalin-fixed, paraffin-embedded tissue slides (FFPE) from seven patients (KRASmut, n=4; KRASwt, n=3), a subset of 97 patients, were strictly prepared for DSP using manual instruction from the GeoMx instrument and the GeoMx immune pathway panel kit with 84 genes (NanoString Technologies Inc.).”

We have added the sentence in the results (lines 254-260 and 351-355).

Lines 254-260 “Based on tissue samples and RNA quality of each sample, different numbers of samples were available for the various assays. We found fresh frozen specimens of patients with KRASwt (n=25) and KRASmut (n=34) were met the RNA quality to study their gene expression profile using the NanoString platform. Notably, between patients (n=59) with KRASwt and KRASmut, we still observed significant overall survival (figure 1B). It implied that next transcriptomic profiling study of 34 patients with KRASmut associated with the worst outcome.”

Lines 351-355 “The seven samples were a subset of 97 patients. CMS classification of seven patients was performed by using their gene expression profiles, which avilable in the Gene Expression Omnibus (GEO, https://www.ncbi.nlm.nih.gov/geo/) repository with an accession number of GSE220148.”

Point 2: The differential expression analyses have not been adjusted (for example, by sex and age), have you considered the use of adjustments?

Response 2: The authors would like to thank the reviewer for this insightful comment. We have not previously considered adjusting for age and sex as these were not statistically different between groups (Table 1). However, we have additionally adjusted for the effect of age and sex for the three genes, FREM1, ERMP1, and CCL8. These are the genes differentially expressed between KRASmut versus KRASwt. The results showed that CCL8 was significantly down regulation in KRASmut after adjusted with sex and age. We have added this finding in line 293.

Line 293 “After adjusted for age and sex, CCL8 still remained statistically differentially expressed lower in KRASmut group (P value=0.01).”

Point 3: I think that this statement should be corrected: “We identified 92 (53 increased and 39 decreased) significant DEG of KRASmut with a P value<0.05, which FREM1, ERMP1 (upregulated) and CCL8 (downregulated) were expressed significantly differently (FDR <0.05)” (lines 214-216). “Significant DEG” and “expressed significantly differently” is the same. 92 genes were nominally significant, while FREM1, ERMP1 and CCL8 were significant after multiple test correction. And this is a key point: since you have tested 770 genes at the same time, you should correct for multiple test. Then, here we have an issue: with only 3 genes actually significant, it does not make sense to do an enrichment analysis. Or, you should justify using the nominally significant genes. In this case, the results showed in 2B, although it is difficult to read, it seems clear that are not significant, since the X-axis is higher than 0.05.

Response 3: The authors would like to thank the reviewer for these valuable comments and suggestions. We have revised the sentence in line 270-274 as “We found 92 (53 increased and 39 decreased) nominally significant DEG of KRASmut with a P value<0.05, which FREM1, ERMP1 (upregulated) and CCL8 (downregulated) were significant after multiple test correction (FDR<0.05).”

The auther have specified that we used nominally significant genes to do an enrichment analysis. We have revised the sentences in line 279-284.

Line 279-284 “We found 92 (53 increased and 39 decreased) nominally significant DEG of KRASmut with a P value<0.05, which FREM1, ERMP1 (up-regulated) and CCL8 (down-regulated) were significant after multiple test correction (FDR<0.05). To gain insight into the pathways involved, we used gene set analysis to determine in which nominally DEG have been annotated or identified using ROSALIND and nCounter software. The analysis of the gene set showed that nominally DEG is enriched in top-five pathways (P value<0.01) in the sources of the BioPanet and WikiPathways database sources (Figure 2B). Surprisingly, we found that nominally DEG were mainly enriched in immune pathways in both databases, such as the immune system pathway (BioPlanet) and the innate immune system (BioPlanet), as well as interactions between immune cells and microRNAs in the TME pathway (WikiPathways).

We have edited the legend of Figure 2B as “differential gene expression between KRASmut and KRASwt with a P value<0.05”. We have changed the X-axis in Figure 2B as P value, and improved the image quality.

Point 4: On the whole, the results are not robust enough to support strong claims like“This study elucidated”, “we demonstrated”, “our study has highlighted”, etc (in abstract, discussion and conclusion). Please, revise those sections to lower the claims.

Response 4: Thank you for suggession. We have revised the words which show strong claims to lower the claims in abstract, discussion and conclusion.

Minor concerns

Point 1: In results, differentially expressed genes have been explained in two different sections (2.6 and 2.9). In which results have been used each one?

Response 1: Thank you the reviewer for comment. We have collaborated section 2.9 with section 2.6 (line). The results of “3.2. DEG of KRASmut tumors enrich in immune signature and TGFb pathways” have been performed by section 2.6.

Point 2: Figures (in the main text and supplementary material) are difficult to see, please provide high quality images.  

Response 2: We apologise for unobvious figures. We have provided high quality images in the main text and suuplementary material.

Point 3: There is not supplemental table S3 (line292)

Response 3: Thank you the reviewer, The author have added supplemeantal table S3 in the supplementary material.

Reviewer 2 Report

Tanjak at al. present a study focusing on KRAS-mutated colorectal carcinoma (CRC). They show the differentially expressed genes in KRAS-mutated CRC, use spatially resolved gene expression profiling to define the effect of KRAS mutation on the tumor micro-evnvironment regions of CRC tissues. They find a specific enrichment of CMS3 CRC in immunosuppressive TME associated with worse prognosis. TGFb signaling activation correlated to reduced pro-inflammatory and cytokine gene signatures leading to suppression of immune infiltration. The study is interesting and well structured, data are clearly presented to support the conlcusions. They provide insights into the characteristics of CMS3 CRC. To improve the research hypothesis should be corroborated by other means, i.e. by measuring protein levels.

Author Response

We thank the reviewer 2 for critically reviewing our manuscript. We are grateful for the reviewers’ constructive and important comments/suggestions to improve the quality of this manuscript. Overall, we have made an attempt to revise the manuscript in accordance to the suggestions of the reviewers. All changes in the manuscript has been shown by “Track Changes” function. We trust that the revised manuscript has been improved to the level of the reviewers’ satisfaction in the Cancers. The point-by-point responses to the reviewer 2’s comments/suggestions/queries are as follows:

Point 1: The differentially expressed genes in KRAS-mutated CRC, use spatially resolved gene expression profiling to define the effect of KRAS mutation on the tumor micro-evnvironment regions of CRC tissues. They find a specific enrichment of CMS3 CRC in immunosuppressive TME associated with worse prognosis. TGFb signaling activation correlated to reduced pro-inflammatory and cytokine gene signatures leading to suppression of immune infiltration. The study is interesting and well structured, data are clearly presented to support the conlcusions. They provide insights into the characteristics of CMS3 CRC. To improve the research hypothesis should be corroborated by other means, i.e. by measuring protein levels..

Response 1: Thank you for precious comments and suggestions. Regarding to reviewer’s suggession, the authors agree to improve the research hypothesis. Currently, we are continually studying the immune cell profiling of CMS3 CRC as a protein level. In this study, we therefore put this point in our limitation in the discussion section Lines 527-535.

Reviewer 3 Report

The KRAS-mutant consensus molecular subtype 3 reveals an immunosuppressive tumor microenvironment in colorectal cancer

In this manuscript, Tanjak et al. carried out a transcriptomic analysis of Kras-mutant colorectal cancer with a focus on the tumor microenvironment (TME). The authors show differential overall survival and gene expression profiles in Kras-mutant vs. Kras-wt colorectal cancer patients. All in all, in line with previous publications and reviews defining Kras-mutant colorectal cancer.

The current manuscript lacks sufficient novelty to be recommended for publication at this point. I would like to add a few comments for the future of this work:

Line 200: It gets confusing when suddenly 59 patients are mentioned without introducing why they were selected among the initial 97 patient cohort. Lines 200-203 should be rephrased.

Fig. 2B: It’s very difficult to read this figure. Please increase the quality of this image.

Fig. 3C: Since the authors want to emphasize that they are presenting novel results with a focus on the TME, it would be more appropriate to display DEG that are specifically occurring in the TME of Kras-mutant tumors in comparison to Kras-wt tumors. Most upregulated genes in the TME compartment are also upregulated on the cancer cells and have been previously described while defining the consensus molecular subtypes.

Furthermore, checking the x axis the changes in gene expression seem to be relatively small. Are you sure the log2 (fold change) data is correct?

Fig. 4: Immune infiltration suppression mediated by TGF-B signaling has been previously described extensively. No novelty here.

Round 2

Reviewer 1 Report

I want to thank the authors for addressing all the concerns, 

Reviewer 3 Report

I would like to thank the authors for clarifying most of the reviewer's concerns. The manuscript has been sufficiently improved.